# Process Simulation and Life Cycle Assessment of Waste Plastics: A Comparison of Pyrolysis and Hydrocracking

**DOI:** 10.3390/molecules27228084

**Published:** 2022-11-21

**Authors:** Muhammad Usman Azam, Akshay Vete, Waheed Afzal

**Affiliations:** School of Engineering, University of Aberdeen, Aberdeen AB24 3UE, UK

**Keywords:** process simulation, waste plastics, pyrolysis, hydrocracking, life cycle assessment

## Abstract

Pyrolysis and hydrocracking of plastic waste can produce valuable products with manageable effects on the environment as compared to landfilling and incineration. This research focused on the process simulation and life cycle assessment of the pyrolysis and hydrocracking of high-density polyethylene. Aspen Plus was used as the simulator and the Peng-Robinson thermodynamic model was employed as a fluid package. Additionally, sensitivity analysis was conducted in order to optimize product distribution. Based on the simulation, the hydrocracking process produced value-added fuels, i.e., gasoline and natural gas. In contrast, pyrolysis generated a significant quantity of pyrolysis oil with a high number of cyclo-compounds and char, which are the least important to be utilized as fuels. Moreover, in the later part of the study, life cycle assessment (LCA) was adopted in order to investigate and quantify their impact upon the environment using simulation inventory data, which facilitates finding a sustainable process. Simapro was used as a tool for LCA of the processes and materials used. The results demonstrate that hydrocracking is a better process in terms of environmental impact in 10 out of the 11 impact categories. Overall, the present study proposed a promising comparison based on energy demands, product distribution, and potential environmental impacts, which will help to improve plastic waste management.

## 1. Introduction

The annual production of plastics has been increasing due to their versatility, light weight, durability, reusability, low cost, and stability. Globally, plastic utilization was just 2 million tonnes in the mid-20th century but elevated to 367 million ton in 2020 [1,2]. Based on the polymer type, polyolefins were the most widely employed polymers, accounting for roughly one-third of total production [1]. However, not only the quality and advantages of plastic materials are imperative to clients; the ecological friendliness of those items is becoming significant too [3]. 

Presently, plastic production accounts for the consumption of roughly 8% of petroleum resources. If the current trends continue, it will utilize 20% by 2050 and consume 15% of the planet’s total carbon budget [4]. Annually, 95% of plastic packaging materials which are worth between USD 80–120 billion, are only utilized for single use and are treated as waste after that [5]. Therefore, the massive production of plastic materials using fossil fuels brings about a serious scarcity of petroleum resources and also collectively contributes to economic loss and significant waste. Figure 1 illustrates an overview of global plastic production data with its projected value and its impact on petroleum consumption and carbon budget. 

In 2015, nearly 302 million tonnes of plastics came to an end as waste. A large share of waste plastic is inadequately managed and is at a high risk of leakage into the natural environment and oceans via waterways and tides, which causes deaths and injuries to countless marine species [6]. Ronkay et al. [7] reported a composition of marine waste with up to 80% plastics, whereas Lebreton et al. [8] discussed waste plastic pollution and reported an annual load of 1.15–2.41 million metric tons of waste plastics in the oceans. Along with environmental impacts, it has been reported that marine plastic wastes and pollution significantly reduce the ecosystem services connected to fisheries and amusement esteem by ~5%, with an annual loss of up to $33,000 per tonne of marine plastic [9].

In order to address waste problems, globally, up to 43% of waste plastics are disposed of in landfills [10]. However, it takes centuries for the degradation of plastic material. Plastics have become a point of major concern due to their non-biodegradability and their presence in waste streams. In developing countries, it has resulted in secondary problems like drain clogging and animal health issues [11]. In parallel, incineration is another primary technique used to address the plastic waste problem by deriving energy from it and simultaneously limiting the quantity of waste disposed of in landfills. It is also considered to be an energy-production facility due to the high calorific value of waste plastics and the lower energy requirements for the operation of plants [12]. However, life cycle reports by Hou et al. [13] and Khoo [14] have demonstrated that notable amounts of particulate matter, greenhouse gases, dioxins, and furans are generated as by-products of this operation. The CO_2_ emissions during incineration alone roughly consume 2% of the overall carbon emissions budget (i.e., 37.5 gigatons in 2018) [15]. 

The recycling of plastics is a substitute technique to landfill disposal and incineration when it comes to energy needs and environmental protection. It secures the ecosystem as well as assists to transform resources into valuable products with minimum utilization of resources [16]. However, contrary to optimistic reports of an improving recycling fraction, only 9% of plastics are recovered annually through recycling [17]. Moreover, recycling entirely relies upon people’s goodwill to sort different waste plastics in recycling bins, while in reality, a significant portion of waste plastics ends up in household waste bags, and it is economically neither productive nor widely employed to recycle a mixture of all types of waste plastics because of its complex nature and due to the presence of different impurities along with the waste plastic material [14]. Similarly, recycling has limited applications because of the loss of plastic properties (i.e., mechanical properties and durability) after recycling it [18]. Merrild et al. [19] also confirmed the ineffectiveness of this technique and claimed that the mechanical recycling of waste plastics results in a 10% average degradation of material along with a 10% quality loss. These issues collectively make it challenging to manage waste plastics through primary or secondary recycling.

Therefore, the current scenario may be defined as a quest for mature yet sustainable approaches that can manage waste plastics with the smallest environmental footprints. Based on the high economic and calorific value of plastics, scientists discovered and have used another favourable route, i.e., chemical recycling, for the management of waste plastics. Chemical recycling involves the cracking of plastics into different chemical intermediates using heat and/or chemicals. These intermediates, typically gases or liquids, are appropriate for use as feedstocks for the production of plastics, petrochemicals, and/or other value-added chemicals [20]. Figure 2 depicts the potential applications of several processing techniques and their associated products for the management of waste plastics. Among all chemical recycling approaches, the pyrolysis and hydrocracking of waste plastics has gained significant consideration due to their mild reaction conditions as compared to gasification and their ability to produce liquid fuels and other value-added products. Pyrolysis includes the break-down of the plastics by heating between 500 and 800 °C in the absence of oxygen in order to yield gaseous and a range of liquid products along with carbonized char, whereas hydrocracking is a chemical recycling process assisted in the presence of hydrogen and utilized for the transformation of bulky plastics with high boiling ranges to saturated hydrocarbons with lower boiling ranges through C–C bond cleavage [16].

Based on the literature, researchers have extensively used different types of plastics, i.e., PET [21], HDPE [22], LDPE [23], PVC [24], PP [25], and PS [26], to generate pyrolysis oil. Moreover, Fivga et al. [27], Almohamadi et al [28], and Selvaganapathy et al. [29] investigated and developed the process simulation model of pyrolysis for the recycling of waste plastics. Khoo [14] modelled an LCA for mixed plastic waste and described pyrolysis as one of the suitable management scenarios which played a significant role in minimizing the environmental impact of mixed plastic waste. Similarly, many researchers have discussed and used various types of catalysts [30,31,32] for the hydrocracking of different waste plastics [33,34] in order to produce liquid fuels and value-added products [35]. However, to the best of our knowledge, no one has discussed and developed a comparative analysis of both recycling methods (i.e., pyrolysis and hydrocracking) based on their process simulation, product analysis, and lifecycle assessment. 

Therefore, the present study presents a comprehensive literature review of both chemical recycling techniques and develops a robust model for the process simulation of pyrolysis and hydrocracking of plastics into liquid fuels, which then be used to perform an industrial-scale computer-aided program to analyse the outputs. HDPE was used as a feedstock because of its extensive waste production and harsh cracking requirements as compared to PP and PS [36]. Based on the analysis of the outputs from both the pyrolysis and hydrocracking processes, extensive product analysis and energy outputs data were developed which were later used as inventory data for the lifecycle assessment of both management scenarios. Sensitivity analysis of the process was conducted in order to optimize the operating conditions. In addition, this study aims to access and compare the potential environmental impacts of HDPE waste in the UK using the LCA method based on ISO standards. 

## 2. Results and Discussion

### 2.1. Overall Material and Energy Balances for Model Validation

A comparison of both hydrocracking and pyrolysis was conducted based on the products formed and lifecycle assessment. The data used for the comparison of the processes were taken from the simulation results. Similarly, the correctness of the process simulation model for hydrocracking and pyrolysis was validated based on mass and energy conservation principles.

#### 2.1.1. Pyrolysis Process

At the end of the pyrolysis process, the HDPE (polyethylene) was broken down into a smaller molecular weight compound. Pyrolysis of HDPE plastic waste at 450 °C and 101.3 kPa produced smaller molecules like methane, ethane, hydrogen, and carbon, etc. The reactor products were passed into a series of coolers and separators as represented in Figure 3.

The main purpose of S-2 is to remove the heavier hydrocarbon in order to get a more refined hydrocarbon fuel. S-3 separated the hydrocarbon fuel into valuable liquids and gaseous products, as seen in streams 10 and 11, respectively. The material balances across the separators with two decimal fractions are given in Table 1 based on the simulation stream results. Moreover, based on the product distribution, the net heating value of the product and their respective enthalpy is part of the table. The mass is balanced across each piece of equipment based on stream data, and the mass across the whole process is conserved. Therefore, the pyrolysis process model abides by the principle of mass conservation. 

The utilities required for the pyrolysis reaction are summarized in Table 2. All utilities (i.e., steam, electricity, cold water, and refrigerant 1) are illustrated in terms of heating and cooling duty. All of the properties of the heating and cooling services are taken from the Aspen plus database. A negative sign indicates heat removed, and a positive sign indicates heat supplied to the equipment.

The overall energy balance of the system was evaluated based on the simulation results, and the data was taken from Table 1 as
Q_Overall_ = H_HDPE_ − H_Products_ + Q_equipment_
Where H_HDPE_ = −1463.7 kWH_Products_ = −274.84 − 37.56 − 323.66 − 92.26 = −733.32 kWQ_equipment_ = 1096.68 + 105.27 − 351.08 − 121.47 = 729.4 kWQ_Overall_ = −1463.7 − (−733.32) + 729.4 ≈ 0

The net heat duty is approximately zero. Therefore, the energy is balanced across the system. Hence, based on the energy and mass balance, the developed process model is validated.

#### 2.1.2. Hydrocracking Process

At the end of the hydrocracking process, the HDPE (polyethylene) was hydrocracked into a smaller molecular-weight compound. The hydrocracking of the HDPE plastic waste was conducted at 375 °C and 6996 kPa of hydrogen pressure in order to produce smaller molecules like methane, ethane, propane, n-butane, iso-butane, pentane, hexane, heptane, n-octane, nonane, decane, undecane, and dodecane. The process of the conversion of HDPE into hydrocarbon fuels took place in multiple stages, as shown in Figure 4.

The material balance for the process with two decimal fractions is given in Table 3 based on the simulation stream results. Moreover, based on the product distribution, the net heating value of the liquid and gaseous products and their respective enthalpies are shown in the Table 3. Based the experimental [37] as well as the simulation results, it was shown that no residual solid or char was formed in the reactor, so S-1 did not separate any solids. Therefore, whatever came from the S-2 stream went out of the S-1 top to S-2 through a series of coolers, as seen in S-3 and S-4. S-2 separated the products into two fractions of gaseous and liquid products, as shown in Table 3. The bottom fraction was liquid hydrocarbons, and the top product was gaseous hydrocarbons, as represented in streams 7 and 8, respectively.

The mass was balanced across each piece of equipment based on the stream data, and the mass across the whole process was conserved. Therefore, the hydrocracking process model followed the principle of mass conservation. All of the utilities (i.e., steam, electricity, cold water, and refrigerant 1) are summarized in Table 4 in terms of heating and cooling duty. All of the properties of the heating and cooling services were taken from the Aspen plus database.

The overall energy balance of the system was evaluated based on the simulation results. The data were taken from Table 3 and calculated as:
Q_Overall_ = H_feed_ − H_Products_ + Q_equipment_
H_Feed_ = H_HDPE_+H_H2_ = −1463.7 + 5.01 = −1458.69 kWH_Products_ = −268.97 − 385.50 = − 654.47 kWQ_equipment_ = 965.71 + 105.27 − 210.02 − 73.97 = 786.99 kWQ_Overall_ = −1461.69 − (−654.47) + 786.99 ≈ 0

The net heat duty (Q_overall_) was approximately zero. Therefore, the energy was balanced across the system, and based on the energy and mass balance, the developed process model was validated.

### 2.2. Sensitivity Analysis

Based on the studies in the literature, sensitivity analysis of both the pyrolysis and hydrocracking processes was conducted in order to optimize the product distribution using ASPEN Plus. Both reactors were considered optimized based on the experimental conditions. However, other equipment needs to be optimized. First, in the pyrolysis process, the optimum temperature of C-1 was obtained by performing a sensitivity analysis of the products formed in the process. On cooling the hydrocarbon products coming from S-1 below 250 °C, an increased amount of S-2 bottom output was observed as shown in Figure 5. A steady product output from S-2 was obtained by further decreasing the temperature. As the temperature dropped below 100 °C, there was a sudden increase in the S-2 bottom products due to the condensation of hydrocarbons below C_8._ Further condensation would cause the carryover of gasoline-range compounds into the heavier compounds. Therefore, 100 °C was considered the optimum temperature for C-2. This ensured the complete removal of the heavy hydrocarbon C_20_^+^ from the product stream. Moreover, the sensitivity for the pyrolysis products (i.e., oils) due to the variation of condenser temperature (C-2) was observed. Interestingly, decreasing the temperature below 70 °C resulted in the condensation of C_5_^+^ compounds. On further cooling, the liquid product obtained from S-3 went on increasing exponentially and was steadied below 25 °C. Therefore, C-2 was set at 20 °C above the ambient temperature in order to obtain a yield of about 65% of the liquid product.

Similarly, the sensitivity analysis and process optimization of hydrocracking were conducted based on the experimental data, and the results are shown in Figure 6. During the process, S-2 was installed in order to separate liquids formed in the process. The optimum temperature of C-2 was obtained by performing a sensitivity analysis of the products formed in the process. As hydrocracking does not produce any solids, the products coming out of S-1 were further cooled in order to remove lower molecular weight compounds from the hydrocarbon mixture. The product was cooled to 15 °C, and this ensured that the gasoline-range products C_5_–C_12_ were condensed and removed. Additionally, the gaseous range products, primarily in the range of natural gas, were also separated out through above-mentioned process.

### 2.3. Simulation Output 

On the basis of the validated models, the results of the simulation were utilized for further analysis and as an input for the lifecycle assessment of the process. Based on the simulation results, the yield of pyrolytic oil and heavy oil was 74.75%, while the yield of synthesis gas was 21.8%. The yield obtained was in accordance with the literature data for PE conversion [29]. The pyrolytic gas obtained through fast pyrolysis produced gaseous products with a heating value ranges around 75.86 MJ/kg, as shown in Table 5. Similarly, pyrolytic oil with a heating value of 44.06 MJ/kg was produced from the pyrolysis of plastic waste. The pyrolytic oil had a boiling point of 139 °C and a flash point of −29 °C based on the simulation results.

Similarly, for the hydrocracking simulation results, the yield of liquid fuel was 41.2%, whereas the yield of synthesis gas was 59.79%. The products obtained from the hydrocracking of HDPE plastic yielded gaseous products with a heating values ranged around 45.82 MJ/kg, as shown in Table 6. This could be used as an alternative to natural gas based on its properties [11]. Similarly, the liquid fuel produced from hydrocracking has gasoline-range properties [11] with a heating value of 44.81 MJ/kg and a boiling point of 69 °C. 

### 2.4. Life Cycle Impact Assessment (LCIA)

Impact assessment is a relative approach based on the functional unit defined in the scope. In the lifecycle impact assessment stage, the overall data are comprehended and evaluated in order to study the environmental impacts of a system. ISO 14044:2006 is normally used to carry out life cycle assessment; it specifies four processes, two of which are mandatory and two others which are optional. The mandatory phases are the selection of impact categories and classifications and characterization. The optional steps are normalization and weighing.

Numerous methods were used for the lifecycle impact assessment in the literature. However, the current study employed SimaPro (8.3.0, multiuser University of Aberdeen 2016) software, and the CML-IA baseline V3.0.4 world 2000 methodology was used for lifecycle impact assessment. The method was developed by the University of Leiden, Netherlands back in 2001. The peculiarity of the method is that it includes more than 1700 different flows which can be downloaded from its website [38]. The CML-IA baseline is a problem-oriented and midpoint approach method by which LCA can be undertaken for a process/product/project, in accordance with the ISO standard regulations. The CML method simulates potential impact assessments of the processes based on 11 impact categories.

Table 7 summarizes the environmental impact results of different impact categories using the CML-IA baseline method. Each of the categories is interpreted individually by comparative analysis of the impact of each contributing factor to both the pyrolysis and hydrocracking scenario. 

#### 2.4.1. Abiotic Depletion

The depletion of abiotic resources is the decrease in the availability of conventional and renewable resources due to improper and unsustainable use [38]. Abiotic depletion can damage natural resources and demolish an entire ecosystem. The depletion of abiotic resources is divided into two types: (i) abiotic depletion of resources which discussed the depletion of elements and ultimate reserves and (ii) abiotic depletion on fossil fuels; which illustrate impact caused by energy utilized in MJ. The overall abiotic depletion results and process contribution to both pyrolysis and hydrocracking processes are shown in Figure 7. Abiotic depletion impact category effects are measured in terms of kilograms of antinomy equivalents (kg Sb eq.). The overall impact values of abiotic depletion on the environment were 0.000143 kg Sb eq. and 0.00017 kg Sb eq. for hydrocracking and pyrolysis, respectively. The overall utility impact in terms of electricity to the processes was 0.000168 kg Sb eq. and 0.000137 kg Sb eq. for pyrolysis and hydrocracking, respectively. The impact of the hydrocracking outputs is as follows. For gasoline fuel it is -0.00023 kg Sb eq., and for natural gas it is −2.9 × 10^−7^ kg Sb eq. On the other hand, outputs from pyrolysis have created an avoided impact of −0.00017 kg Sb eq. for pyrolysis oil and −1.1 × 10^−7^ kg Sb eq. for gaseous products. Similarly, coke and heavy oils produced in the process have an avoided impact of −3.1 × 10^−5^ and −2.5 × 10^−5^ kg Sb eq. on the environment, respectively. Overall, the major difference in the effect on abiotic depletion is due to the difference in the utility required for each of the processes. Pyrolysis requires a higher amount of utility which in turn creates a higher potential impact on the environment. Overall, both pyrolysis and hydrocracking showed a negative impact on abiotic depletion with positive values for abiotic depletion.

Similarly, the overall abiotic depletion results for both the pyrolysis and hydrocracking processes are compared in Figure 8. The abiotic depletion impact category effects are measured in terms of the amount of energy utilized, i.e., MJ. The overall impact of the depletion of fossil fuels for pyrolysis is 29,901.52 MJ, and for hydrocracking it is 24,528.46 MJ. The impact created with the use of HDPE as the feed is equal. The input utility used in pyrolysis creates an impact of 11,915.8 MJ, which is more than 1.2 times that of the hydrocracking process. The oil generated in the pyrolysis process creates a positive impact of −36,652.54 MJ as compared to gasoline fuel, −24,380.7 MJ in the hydrocracking process. The major difference in the results of the two processes is due to the lower contribution of pyrolysis products as compared to hydrocracking. Overall, pyrolysis produces a greater impact on the environment in the category of fossil fuel depletion.

#### 2.4.2. Global Warming Potential (GWP)

Climate change is the potential group of global warming potential, and it refers to the variation in the global temperature caused by the release of greenhouse gases [39]. Global warming potential (GWP) is used in LCA to measure greenhouse gas emissions, which increase the natural greenhouse effect due to the adsorption of Earth’s emissions [40]. It works as an impact indicator which measures the changes in the global temperature and climatic conditions [38]. Figure 9 shows the overall climate change results for the pyrolysis and hydrocracking of HDPE. The impact category of climate change or the global warming potential is measured in kg CO_2_ equivalents (kg CO_2_ eq.). In the case of global warming potential, pyrolysis has an impact of 2557.997 kg CO_2_ eq., and hydrocracking has an impact of 1049.399 kg CO_2_ eq. The input used in both of these processes is the same, so the GWP impact is nearly similar with a slightly higher impact from the hydrocracking process due to the presence of hydrogen in the feed. The utility input in terms of electricity used has a slightly lower impact on global warming in the case of hydrocracking as compared to pyrolysis, with the values of 835.2 kg CO_2_ eq. and 1019.2 kg CO_2_ eq., respectively. For hydrocracking, overall GWP is benefitted by a higher positive impact caused by the production of gasoline-range products which is 5.17 times the impact caused by the production of pyrolysis oil. In conclusion, hydrocracking shows an overall lower climate change impact on the environment as compared to pyrolysis.

#### 2.4.3. Ozone Layer Depletion (ODP)

The ozone layer is the protective layer in the stratosphere. The depletion of this protective layer by ozone-depleting substances like gases is called ozone layer depletion. It is indicated by the increase in UV-B radiation and subsequent rise in skin illness among humans. The overall results of ozone layer depletion and input and output contributions to ozone depletion are summarized in Figure 10. The units used to express the impact on ozone layer depletion in the environment are kg CFC-11 eq.

Overall, pyrolysis has an impact of −0.00044 kg CFC-11 eq., and hydrocracking has an impact of −0.00023 kg CFC-11 eq. The contribution of utility to the overall results is slightly higher in the case of pyrolysis due to higher utility usage. Additionally, the contribution of pyrolysis products would require the process to burn fossil fuels, which would affect the ozone layer and contributes positively to the environment. Gasoline-range fuels produced during the hydrocracking process have a value of −0.00029 kg CFC-11 eq. as compared to −0.00044 kg CFC-11 eq. of pyrolysis oil. Pyrolysis oil produced in the process plays a deciding factor in defining the overall impact of the two processes. The overall effect of both of the processes on ozone depletion is nearly the same. However, numerically, pyrolysis has a greater positive impact on ozone layer depletion potential than the hydrocracking of HDPE feed.

#### 2.4.4. Human Toxicity

Human toxicity potential is a direct indicator that denotes the effect of toxic chemicals on humans. Human toxicity potential is defined as the potential effect a toxic chemical can have upon human life. A high toxicity potential can cause life-threatening diseases like cancer in humans. The overall results of human toxicity and individual contributions from the contributing factors are indicated in Figure 11. Human toxicity is measured in terms of kg 1,4 dichlorobenzene equivalent (kg 1,4-DB eq.). The overall results of the effect on humans in terms of human toxicity are 208.9515 kg 1,4-DB eq. and −7194.09 kg 1,4-DB eq. for the pyrolysis and hydrocracking processes, respectively. Gasoline-range fuels and natural gas produced in the hydrocracking process contribute about −7476.35 kg 1,4-DB and −1.73513 kg 1,4-DB eq. to human toxicity, respectively, while the oil and gas produced during pyrolysis contribute −91.9663 kg 1,4-DB eq. and −0.634 kg 1,4-DB eq., respectively. Hydrocarbon fuel outputs contribute positively to the environment, as they reduce the impact caused when producing the same products from conventional sources. The input values for human toxicity are higher for pyrolysis, as the utility requirement is higher in that case. In the case of hydrocracking, the overall value of human toxicity is benefited by the production of gasoline fuel in the process. Overall, there is a huge gap in the results of human toxicity when pyrolysis and hydrocracking are compared, and a viable option (i.e., hydrocracking) is present for the management of waste plastics after considering its beneficial results on human toxicity.

#### 2.4.5. Fresh Water Ecotoxicity (FAETP)

The effect of emissions or chemicals on the ecosystem is known as ecotoxicity. Ecotoxicity is measured as the impact of toxic substances on freshwater, marine, and terrestrial ecosystems. The effects of ecotoxicity are loss of biodiversity and extinction of species, which in turn damage the quality of the ecosystem. Figure 12 shows the impacts of hydrocracking and pyrolysis and the individual contributions of inputs and outputs to freshwater ecology. The unit used for measuring fresh water ecotoxicity is kg 1,4 dichlorobenzene equivalent (kg 1,4-DB eq.).

The overall freshwater ecotoxicity results for hydrocracking are 188.860 kg 1,4-DB eq., and those for pyrolysis are 204.0771 kg 1,4-DB eq. The individual contribution of feed to the processes is around 73 kg 1,4-DB eq., with a slightly higher value in the case of hydrocracking due to the presence of hydrogen in the feed. The freshwater ecotoxicity value of the utility input in the form of electricity is higher for the pyrolysis of HDPE feed due to the higher utility requirement. Pyrolysis process contributions a positive impact of −42.9163 kg 1,4-DB eq. from oil, −6.245 kg 1,4-DB eq. from heavy hydrocarbons, and −0.007 kg 1,4-DB eq. from gas. Similarly, gasoline fuel contributes about −34.0578 kg 1,4-DB eq., and gaseous product contributes about −0.00191 kg 1,4-DB eq. in the case of the hydrocracking process. The overall impact of both processes on freshwater ecology is negative. Both processes show a minimal difference on fresh water ecotoxicity. However, in case of pyrolysis, the impact is higher due to the higher utility requirement for the maintenance of a relatively higher temperature in the reactor.

#### 2.4.6. Marine Aquatic Ecotoxicity (MAETP)

Figure 13 shows the overall and individual contributions to marine aquatic ecotoxicity of the pyrolysis and hydrocracking of HDPE waste. The impact of marine aquatic toxicity is measured in terms of kg 1,4 dichlorobenzene equivalent (kg 1,4-DB eq.). Overall, both pyrolysis and hydrocracking showed a negative impact on marine aquatic ecotoxicity with a value of 1,309,682 kg 1,4-DB eq. and 1,156,427 kg 1,4-DB eq., respectively. The marine ecotoxicity contributions of the inputs of the hydrocracking process are 359,359 kg 1,4-DB eq from feed and 945,357.6 kg 1,4-DB eq. from utility inputs. On the other hand, the contributions of the pyrolysis process are 358,565.9 kg 1,4-DB eq. from feed and 1,153,622.9 kg 1,4-DB eq. from utility input. In terms of avoided products, hydrocracking products showed a positive impact of −16,243.1 kg 1,4-DB eq. and −132,047 kg 1,4-DB eq. for natural gas and gasoline fuel respectively whereas, pyrolysis products showed a positive impact of −165,002 kg 1,4-DB eq. for oils, and −5941.07 kg 1,4-DB eq. for synthesis gas. The high utilization of electricity in pyrolysis creates the major difference between the ecotoxicity effects of hydrocracking and pyrolysis on marine ecology. Electricity creates the maximum negative impact on the marine ecosystem. In conclusion, hydrocracking fairs better as compared to pyrolysis when the environmental impact on marine ecology is considered.

#### 2.4.7. Terrestrial Ecotoxicity (TAETP)

Figure 14 shows the results of the environmental impacts on terrestrial ecotoxicity and the contributing factors in hydrocracking and pyrolysis of plastic feed. Terrestrial ecotoxicity is measured in units of kg 1,4 dichlorobenzene equivalent (kg 1,4-DB eq.). The overall terrestrial ecotoxicity results for hydrocracking are −0.31712 kg 1,4-DB eq., while those of pyrolysis are 0.6203 kg 1,4-DB eq. The individual contributions of feed to the processes are around 0.255 kg 1,4-DB eq. with slightly higher values in case of hydrocracking due to the presence of hydrogen in the feed. The terrestrial ecotoxicity value of the utility input in the form of electricity is higher for the pyrolysis of HDPE feed due to the higher utility requirement. In the case of hydrocracking, products showed a positive impact of −0.00316 kg 1,4-DB eq. for natural gas and −1.44141 kg 1,4-DB eq. for oils, while for pyrolysis products, it valued −0.56392 kg 1,4-DB eq. for oil and −0.00116 kg 1,4-DB eq. for gaseous products. Hydrocracking is benefitted by its lower utility requirement. Overall, hydrocracking creates a positive impact on the terrestrial ecology with a negative value; however, pyrolysis creates a negative impact on the terrestrial ecosystem.

#### 2.4.8. Photochemical Oxidation Potential

Photochemical oxidation is also known as photochemical ozone creation potential. Photochemical ozone or ground-level ozone is formed when volatile organic compounds (VOCs) react with NOx in the presence of heat and sunlight. It affects the quality of ecosystems and human health. Smog in the atmosphere acts as an indicator of photochemical oxidation. The overall results of photochemical oxidation and individual contributions from the contributing factors are shown in Figure 15. Photochemical oxidation is measured in terms of kg C_2_H_4_ eq. The overall results of photochemical oxidation are 0.2814 kg C_2_H_4_ eq. for the pyrolysis process and −7.25 kg C_2_H_4_ eq. for the hydrocracking process. Hydrocracking gas and gasoline-range fuels produced during the hydrocracking process contribute to the avoidance of −0.08383 kg C_2_H_4_ eq. and −8.00329 kg C_2_H_4_ eq. towards photochemical oxidation, respectively, while oil and gas produced during pyrolysis avoided −0.215 kg C_2_H_4_ eq. and −0.0306 kg C_2_H_4_ eq., respectively. The hydrocarbon fuel outputs contribute positively to the environment, as they reduce the impact caused by producing the same products from conventional sources. The contribution of the utility sources to photochemical oxidation is nearly similar for both processes. Overall, there is a huge difference in the results of photochemical oxidation when pyrolysis and hydrocracking are compared.

#### 2.4.9. Acidification Potential

Acidification is the reduction in pH due to the acidifying effect of gaseous emissions. Gases like SO_x_, NO_x_, and ammonia cause acidification of the ecosystem and affect biodiversity. The impact of acidification can be seen in increased acidity in water and other ecosystems. Figure 16 shows the results of the environmental impact of acidification and its contributing factors from the hydrocracking and pyrolysis of HDPE. The acidification is expressed in units of kg SO_2_ eq.

The overall results of acidification are 3.59 kg SO_2_ eq. in the case of hydrocracking and 7.82 kg SO_2_ eq. in the case of pyrolysis. The feed inlet for both processes contributes almost equally with a slightly lower contribution from pyrolysis due to the presence of hydrogen in the hydrocracking feed. The utility contribution to acidification follows a similar trend as in the other impact categories. The major liquid fuel in both cases contributes to provide a positive impact to the environment, and the same is true in the case of the gaseous products in both processes. The heavy hydrocarbons formed in pyrolysis have acidification value of −0.508 kg SO_2_ eq. In conclusion, the overall results of the impact in the acidification category are a result of the contributions of individual contributors with no single major contributor creating a difference in the environmental impact created by the two processes. Overall, the results show that both of the process have the negative effect on acidification. However, hydrocracking valued lower impact as compared to pyrolysis. 

#### 2.4.10. Eutrophication Potential 

Eutrophication is the accumulation of nutrients in the ecosystem. The impacts of eutrophication are a rise in the concentration of phosphorus and nitrogen in aquatic systems and formation of algae. The increase in the concentration of nutrients or biomass formation in the ecosystem affects its quality. Figure 17 shows the overall eutrophication results for the pyrolysis and hydrocracking of HDPE waste, including the process contribution to the total potential. The impact category of eutrophication is measured in kg phosphate equivalents (kg PO_4_ eq.). In the case of eutrophication potential, overall, pyrolysis has an impact of 1.336 kg PO_4_ eq., and hydrocracking has an impact of 0.237 kg PO_4_ eq. The overall utility input in terms of electricity used has a lower impact on eutrophication in the case of hydrocracking as compared to pyrolysis, with the values of 1.02 kg PO_4_ eq. and 1.24 kg PO_4_ eq., respectively. The eutrophication potential for gasoline fuel in hydrocracking is much higher than the liquid fuel formed in pyrolysis. Therefore, hydrocracking shows an overall lower eutrophication impact on the environment as compared to pyrolysis.

## 3. Materials and Methods

The advantages of the chemical recycling of plastics, i.e., pyrolysis and hydrocracking, over other mechanical or energy recovery methods are obvious. However, a detailed comparative analysis between these two chemical recycling methods needs to be quantified based on process modelling and simulation, product analysis, and lifecycle assessment. Therefore, the current section expressed in detail an extensive literature review of both the pyrolysis and hydrocracking methods in order to compare the process simulation along with their lifecycle assessments. 

### 3.1. Comparative Analysis of Pyrolysis and Hydrocracking

Thermal cracking or pyrolysis includes the break-down of the plastics by heating between 500 and 800 °C and in the absence of oxygen in order to yield gaseous and a range of liquid products along with carbonized char [41]. In terms of reaction conditions, the pyrolysis of waste plastics can be carried out in three different temperature ranges: low temperature (<500 °C), medium temperature (500–800 °C), and high temperature (>800 °C) pyrolysis [42,43]. Various kinds of catalysts are used to improve the pyrolysis process and its efficiency. Optimizing the operating conditions helps to reduce the temperature and target specific products. However, the extent of each fraction and their exact composition depends essentially upon the reaction conditions together with the catalyst and polymer type. 

Typically, zeolites, FCC, and mesoporous materials, e.g., MCM-41, are used for catalytic cracking [44,45]. Lee et al. [46] studied the cracking of HDPE at 400 °C using spent FCC catalyst and yielded a total conversion of 98% with a maximum selectivity of liquids towards olefins, paraffins, and aromatics. Similar results were obtained by Abbas-Abadi et al. [47], who studied the impact of temperature on the product yield and distribution of the catalytic pyrolysis of HDPE at a temperature range of 420–510 °C using FCC as a catalyst. The maximum yield of condensed products was achieved at 450 °C with a selectivity of 73.78% for olefins. An Increase in the reaction temperature resulted in an increased content of gaseous products and coke. This was due to the conversion of condensed products into aromatics and non-condensed gases and finally char. Similarly, in terms of carbon number distribution, 78.42% of the products belonged to C_5_–C_9_. In another study by Miskolczi et al. [48], the researchers investigated the thermo-catalytic behaviour of HDPE at 450 °C. The addition of catalyst significantly altered the composition of products. Similarly, HZSM-5 yielded the maximum gaseous products with minimum residue, whereas the yield of liquids follows the flowing trend: FCC > HZSM-5 > NCM > thermal. This was due to the difference in the microporosity, mesoporosity, and acidity of the catalysts. Moreover, Mastral et al. [49] studied the impact of temperature and residence time on the pyrolysis of HDPE in a fluidized reactor. Lower temperature resulted in waxy products, while an increase in temperature drastically changed the product distribution to gaseous products. Similarly, time played a key role as a slight increase in residence time shifted the product distribution from liquids to gases. A detailed summary of HDPE waste pyrolysis studies is illustrated in Table 8.

Similarly, hydrocracking is another type of cracking in which complex feedstocks like plastics, heavy oils, etc., are converted into lower boiling compounds such as diesel, gasoline, and kerosene. It is a process wherein the hydrogenation of aromatics and cracking occur simultaneously. It is carried out at elevated pressure and temperature in a hydrogen-rich environment [50]. Generally, a bifunctional catalyst is required to perform a hydrocracking reaction in a stirred batch autoclave reactor with the temperature and cold hydrogen pressure ranges of 300–450 °C and 2–15 MPa, respectively [16,51]. Heat energy is needed to achieve the desired temperature for reaction contents and to break the long chain polymers, while cold hydrogen pressure in the reaction medium should be utilized to overcome repolymerization and dehydrogenation reaction, which eventually promotes coke formation [52]. From the energy perspective, both hydrogenation and cracking are complementary reactions, as the former is exothermic while the latter represents an endothermic reaction [53]. 

Up to now, researchers have discussed and used various types commercially available and in-house-fabricated catalysts with different acidic and metallic supports for the hydrocracking reaction. Generally, noble metals such as Pt or Pd [33,54] and transition metals such as Mo, Ni, W, and Co [30,31,55] are employed as metals impregnated over acidic support, whereas microporous zeolites such as HZSM-5, HUSY, Zβ [30,31,32], hierarchical zeolites [56], sulfated zirconia [33,34], natural clays (e.g., bentonite montmorillonite, etc.) [57], and silica-alumina [35], etc., have been utilized as a source of acidic support. Recently, Costa [30] employed H-USY and H-ZSM-5 zeolites for the hydrocracking of HDPE. The reaction was performed at 390 °C for 60 min with a feed-to-catalyst ratio of 9:1. Both of the catalysts showed a 100% conversion rate. However, H-USY was more selective towards lighter hydrocarbons (i.e., C_8_–C_12_ and C_13_–C_20_), whereas H-ZSM-5 showed selectivity towards higher hydrocarbons (i.e., C_21_–C_38_). This was due to the higher porosity and external surface area of H-USY as compared to H-ZSM-5. The author proposed a high cracking rate on the external surface of H-USY (i.e., 189 m^2^ g^−1^) which lead to the formation of lighter hydrocarbons, whereas diffusion limitations of the molecules within the small pores of H-ZSM-5 (i.e., V_micro_ = 0.13 cm^3^ g^−1^) resulted in a low-cracking and higher hydrocarbon product. 

Moreover, Pan et al. [58] investigated the beneficial behaviour of Ni metal in terms of enhancing the aromatization capacity of HZSM-5 for the hydrocracking of HDPE. With the addition of 10 wt % Ni, the selectivity of alkanes increased from 33.0% to 41.1%, and a further addition to 20 wt% resulted in a drop in selectivity to 32.9%. However, a reverse trend was seen for aromatics selectivity, which first decreased (i.e., from 0 to 10 wt% of Ni) and then increased (i.e., from 10 to 20 wt% of Ni), while the selectivity of olefins gradually dropped. The authors suggested the increase of aromatics at the expense of alkanes and olefins. Similarly, the authors explained that the decrease in the selectivity of aromatics from 0 wt% to 10 wt% of metal was due to the behaviour of Ni, which promotes alkylation and inhibits the cyclization of olefins in order to prevent further de-hydro aromatization. Interestingly, a further increase in Ni content aided the aromatization of olefins and justified the beneficial behaviour of Ni towards the aromatization of olefins.

**Table 8 molecules-27-08084-t008:** Summary of HDPE pyrolysis studies.

Catalyst	Reactor	F/C *	T (°C)	Gas Mass %	Oils Mass %	Residue Mass %	Comments	Ref
Spent FCC	Semi-batch	10:1	400	16	82	2	Spent FCC catalyst as supplied by SK Co. Ltd was utilized for the pyrolysis of HDPE at 400 °C. The results showed around 80% of olefins in the oily products due to the catalytic degradation. However, these olefinic intermediates were unchangeable to paraffins by the hydrogenation reaction.	[46]
FCC **	Semi-batch	5:1	420	6.7	89.1	4.2	HDPE degradation experiment was performed in a 1 L reactor at different reaction temperatures using FCC as a catalyst. The optimized reaction temperature with minimal coke formation was 420 °C. At this temperature, 73% of the liquids showed olefinic composition with a maximum composition of C_5_–C_9_ products.	[47]
			450	4.7	91.2	4.7		
			480	8.8	85.3	5.9		
			510	12.9	79.5	7.6		
Thermal/No catalyst	Batch	-	450	5.8	74.5	19.7	The pyrolysis of HDPE was studied at 450 °C using a range of catalysts. As compared to thermal run, all the catalyst showed significant decomposition. Also, the addition of catalyst narrower to the carbon number distribution of liquids along with an increase in yield of gasoline and kerosene-range products. The effect of catalysts followed the decreasing trend of ZSM-5 > FCC > NCM.	[48]
NCM ***		33:1		6.3	78.5	15.2		
FCC				6.3	82.5	11.2		
ZSM-5				15.1	81	3.9		
Thermal/No catalyst	Mini bench top reactor	-	500	7	93	0	100% cracking of HDPE was thermally conducted at 500 °C for 1 h. Gaseous products mainly contained methane, ethane, propane, and butane. Similarly, oils showed maximum selectivity towards aromatics.	[59]
Thermal/No catalyst	Batch	-	440	17	74	9	Thermal pyrolysis of grocery bags followed by fractional distillation was conducted in order to produce diesel-range fuels with a maximum selectivity of aliphatic paraffinic hydrogens and a small amount of aliphatic olefinic hydrogens and aromatic hydrogens.	[60]
HZSM-5	Conical spouted bed	30:1	500	58	41.93	0.06	Catalytic pyrolysis of HDPE was conducted using zeolite-based catalysts at 500 °C. HZSM-5 produced a large quantity of light olefins because of the small pore size of the material, whereas Hβ showed a maximum yield due to its larger pore size as compared to HY and HZSM-5.	[44]
HY				20	79.8	0.2		
Hβ				25	74.8	0.2		
Hβ	Batch Reactor	100:1	380	9.28	45.1	45.7	Hierarchical Hβ was prepared and utilized for the cracking of HDPE at mild conditions. Compared to the commercial Hβ, hierarchical Hβ showed better conversion with a maximum productivity of gasoline in olefins-range fuels.	[45]
Hβ (CTAB)				17.02	50.27	32.7		
Hβ (PHAPTMS)				15.13	81.86	3		
Al-MCM-41				-	-	>95		
Thermal/No catalyst	Fluidized Bed	-	650	31.5	68.5	-	The pyrolysis of HDPE was investigated, and the impact of temperature and residence time on the product distribution was examined. An elevation in the reaction temperature significantly shifted the oils to gaseous products. Similarly, an increase in residence time had an influence on the gas composition, and it became more significant as the reaction temperature increased.	[49]
			650	22.1	72.3	-		
			780	78.8	15.3	-		
			780	85.6	9.6	-		
			850	75.1	11.4	-		
			850	64.5	12.2	-		

* Feed to catalyst ratio; ** A silica alumina catalyst with Si/Al ratio of 6; *** Clinoptilolite having rhyolite tuff.

Zhang and co-workers [58] observed the effect of catalyst-to-feed ratios of 0.02, 0.04, 0.06, and 0.08 on the hydro-liquefication of HDPE using Ni-HZSM-5 at 400 °C for 2 h. The aromatics selectivity enhanced linearly with the increase in catalyst amount, and it reached a maximum of ~14% at a catalyst: feed ratio of 0.08, whereas olefins and alkanes showed a decrease in selectivity from 16.3% to 4.5% and 34.6% to 24.8%, respectively. The author explained this effect based on two factors: (i) the addition of catalyst encouraged the breakage of long-chain hydrocarbons in order to yield more olefins which were favourable to the production of aromatics; (ii) additional HZSM-5 offered more sites for olefin cyclization and dehydrogenation. A detailed summary of the literature based on previous studies is presented in Table 9.

Only few scientists have explored the impact of hydrogen pressure on the hydrocracking of waste plastic. Ding et al. [37] studied hydrocracking at two different hydrogen pressures and observed a significant influence of cold hydrogen pressures initially. However, the effect was decreased with the further increase in pressure. In detail, the increase of hydrogen pressure from 1.83 MPa to 5.27 MPa changed the conversion from 84.9% to 98.9%. Contrary to Ding et al. [37], Luo et al. [61] observed the effect of hydrogen pressure on the hydrocracking of mixed plastic. With an increase of pressure from 2.3 MPa to 8.6 MPa, the conversion as well as yield of oil first decreased and then later increased, whereas the yield of gas remained unaffected. In summary, research has shown that hydrogen pressure does have an impact on conversion and liquid yield, but that this impact decreases with increasing hydrogen pressure. Higher hydrogen pressures, however, increase the quality of the fuel produced. Coke formation and the removal of contaminants from waste plastics may both benefit from increasing hydrogen pressure. Therefore, it can be concluded that the best cold hydrogen pressure for the hydrocracking of waste polymers is between 2.0 and 6.0 MPa.

Similarly, Costa et al., [30] studied the effect of reaction temperature on liquid product distribution for the hydrocracking of HDPE using H-USY zeolite. Increasing the temperature from 325 °C to 390 °C, significantly enhanced the product distribution from heavy hydrocarbons (i.e., C > 21) to gasoline (C_5_–C_12_) and diesel-range (C_13_–C_20_) products. The author concluded that increasing the temperature from 325 °C to 390 °C accelerated the degradation of HDPE, which resulted in the formation of short-chain hydrocarbons. In the same group, Silva, and co-authors [30] examined the temperature-dependency of product selectivity for the hydrocracking of HDPE under H-ZSM-5. As expected, the heavier hydrocarbons (i.e., C > 21) were reduced from 68 to 64%, while the gasoline fraction was increased by an augmentation of the reaction temperature from 360 °C to 390 °C. As a result, higher temperatures potentially promote products’ selectivity towards lower molecular weight.

Overall, both the hydrocracking and pyrolysis of HDPE and the formation of desired products depend upon several parameters, i.e., temperature, catalyst employed, feed: catalyst ratio, time, hydrogen pressure, etc., and therefore it is difficult to form any conclusion without performing an experiment. However, based on the literature, a process simulation model could be developed which helps us to understand the processes in detail. Also, based on its validation, the process could be utilized for the analysis of any kind of waste plastic.

**Table 9 molecules-27-08084-t009:** Summary of HDPE hydrocracking studies.

Process/Catalyst	T (°C)	t (min)	P_H2_ ¤ (MPa)	F/C **	Gas Mass %	Oils Mass %	Liquids Mass %	Conversion Mass %	Summary	Ref
No catalyst/Thermal	375	60	7	-	0.17	2.22	-	2.39	Addition of bifunctional catalysts significantly increased the conversion of HDPE cracking. Similarly, Ni/HSiAl showed the maximum conversion due to its metal sulfide–acid balance. An increase in catalyst loading showed enhanced conversion with a notable increase in gas yields. In terms of product distribution, both KC-2600 and Ni/HSiAl produced better-quality liquids with more iso-paraffins and less aromatics.	[37]
KC-2600 *	375	60	7	1.5:1	57.2	32.8		90		
HSiAl	375	60	7	1.5:1	-	-	-	66		
Ni/HSiAl	375	60	7	1.5:1	57.6	42		99.6		
NiMo/HSiAl	375	60	7	1.5:1	54.1	45.2		99.3		
KC-2600	375	60	7	4:1	24.4	40.1		64.5		
Ni/HSiAl	375	60	7	4:1	38.9	42.8		81.7		
NiMo/HSiAl	375	60	7	4:1	22.7	42.6		65.3		
No catalyst/Thermal	400	60	5	-	12.9	-	86.6	100	At lower temperature, HDPE was converted to waxy compounds. An increase in temperature significantly enhanced the gas yield with a decrease in the liquid products. Temperature was maintained at 425 °C and considered as the optimal temperature for the hydro-liquefication of HDPE. The naphtha obtained over HYDROBON showed a low olefin content, whereas DHC-8 has a high olefin content and should be hydrogenated before use.	[62]
No catalyst/Thermal	425	60	5	-	17.0	-	81.5	98.9		
No catalyst/Thermal	450	60	5	-	18.4	-	77.6	96.5		
DHC-8 ***	400	60	5	20:1	10.3	-	85.5	96.8		
HYDROBON	400	60	5	20:1	4	-	93.1	98.1		
50% DHC-8 + 50% HYDROBON	400	60	5	20:1	10.5	-	87.0	98		
DHC-8	425	60	5	20:1	19.0	-	79.5	97.8		
HYDROBON	425	60	5	20:1	13.5	-	85.8	99.9		
50% DHC-8 + 50% HYDROBON	425	60	5	20:1	18.9	-	80.2	99.7		
DHC-8	450	60	5	20:1	26.7	-	67.1	94.8		
HYDROBON	450	60	5	20:1	20.5	-	76.9	96.9		
DHC-32	430	60	8.3	-	8.5		35.2	43.6	ZSM-5 showed the maximum conversion and productivity of liquids due to high acidic characteristics and the surface area of the catalyst.	[63]
FCC ****	430	60	8.3	-	7.5		37.5	45		
NiMo/γ-Al_2_O_3_	430	60	8.3	-	9.6		40.2	50		
ZSM-5	430	60	8.3	-	7.2		50.4	57.3		
BC27	400	60	2	20:1	32		68	98	Both mesoporous BC27 and BC48 catalysts showed a maximum conversion of 98% with a liquid yield above 68 wt.%. In terms of product distribution, both catalysts exhibited a maximum selectivity towards gasoline-range fuels. This was because of the high external S_A_ of the synthesized catalysts.	[64]
BC48	400	60	2	20:1	38		68	98		
Low Alumina FCC (with solvent)	440	30	5.6	4:1	6.2		74.1	80.3	The addition of solvent significantly affected the cracking of HDPE with both FCC and zeolite-based catalysts. An increase in the reaction temperature to 440 °C notably promoted the overall conversion. However, product distribution followed the gaseous-range fuels.	[61]
Low Alumina FCC (without solvent)	440	30	5.6	4:1	11.1		82.2	93.3		
HZSM-5 (with solvent)	400	30	5.6	4:1	18.3		17.9	36.1		
HZSM-5 (with solvent)	440	30	5.6	4:1	61.5		37.9	99.4		
No catalyst/Thermal	500	60	1		5		95.0	100	A 100% hydro-liquification of HDPE was achieved at 500 °C with an enhanced productivity of oily products with a significant amount of double-ring aromatics.	[52]

¤ Cold hydrogen pressure; * May contain NiMo/zeolite and/or NiMo/Al_2_O_3_; ** Feed-to-catalyst ratio; *** DHC-8 is an amorphous catalyst consisting of non-noble metals on a SiAl base; **** Metal supported on zeolite Y.

### 3.2. Process Simulation

Process modelling makes it easier to study and investigate any process, as it represents a real-life process. Simulation helps in studying the effect of various process parameters and investigating the factors affecting the quality, yield, and selectivity of products. Aspen Plus was used to simulate both the pyrolysis and hydrocracking processes, as it has a huge library of components, equation of state (EOS), and block models for various unit operations, which reduces its reliability on other resources. Implementing the model-based simulations makes it easier to study the mass and energy balance for the process and also to determine the cost and size of the equipment. Both of the processes were modelled at a steady state, and it was assumed that the reactor is adiabatic and that all of the feed (i.e., HDPE) was reacted. The processes were modelled and simulated using Aspen Plus V12.1 software.

Pyrolysis and hydrocracking were modelled for comparative analysis and utilized HDPE plastic waste as the main feed with a flow rate of 1000 kg/h. The feed conditions were set to atmospheric conditions (i.e., 25 °C), and the feed physical properties (density 967 kg/m^3^) for HDPE were taken from the Aspen Plus database. The reactor conditions were set at 450 °C and 101.3 kPa for pyrolysis and 375 °C and 6996 kPa for hydrocracking based on studies by Selvaganapathy et al. [29] and Ding et al. [37], respectively. The feedstock used in the simulation process is summarized in Table 10.

#### 3.2.1. Pyrolysis

The simulation uses Peng-Robinson (PR) as the thermodynamic package, as it provides accurate results for lighter gases like hydrogen based on the Aspen Plus resources. Moreover, in a similar study, Adeniyi et al. [65] also investigated the pyrolysis of waste plastics using Peng-Robinson (PR) as the property method. Polyethylene (PE) was used from the Aspen Plus databank, treating it as a conventional component, while its properties and all the other component properties were generated using the Aspen Plus property generator for the simulation process. The products from the reactor, which are mixture of different hydrocarbons based on simulation study, were treated as conventional components from the database

The simulation environment was set up based on the process flow as illustrated in Figure 16. The flow diagram was prepared using the unit operation blocks and parts from the model palette. The HDPE plastic waste feed was first pre-heated in a heater (H) and was further transferred to a high-temperature reactor (450 °C and 101.3 kPa) in an inert environment in order to produce pyrolysis oil, gas, and char. The pyrolysis reactor was simulated using a reactor yield model in Aspen Plus. The reactor was set up based on the simulation results of Selvaganapathy et al. [29]. The pyrolysis reaction is complex, and the products are uncertain, which makes it difficult to obtain accurate reaction kinetics. Therefore, the reaction was simulated using the RYield block. It is appropriate to use the RYield reactor, as the reaction is irreversible, and the products produced are considered fast pyrolysis. The main product obtained was liquid hydrocarbons. 

Further, a cooler separator combination was used in order to obtain three different cuts of hydrocarbon, as shown in Figure 1. The stream from the reactor was first separated in S-1 in order to obtain solid char, while the rest of the stream was passed through cooler C-1 to cool it to 100 °C. After the first cooler, the stream entered the separator S-2 to obtain the first cut of hydrocarbon fuel. The top product from S-2 was passed through cooler C-2 (20 °C), and the cooled stream was separated in the final separator (S-3) in order to obtain the top and bottom gas and liquid products. The models selected along with operational parameters and design specifications used for modelling the process were summarised in the Table 11. 

#### 3.2.2. Hydrocracking

In the case of hydrocracking, the fluid package was selected for physical property estimation. Peng-Robinson (PR) was used as the thermodynamic package, because it provides accurate results for lighter gases like hydrogen based on the Aspen Plus resources, and hydrogen is the major feed component in hydrocracking. For hydrocracking, feed polyethylene (PE) and hydrogen were used from the Aspen Plus databank, and both were treated as conventional components. The feed conditions were kept constant in order to conduct a comparative analysis of the processes. Similar to pyrolysis, the products from the reactor, which were a mixture of different hydrocarbons based on a simulation study, were also considered as conventional components.

The simulation environment was built based on the process flow diagram of hydrocracking, as shown in Figure 4 using Aspen Plus. HDPE plastic waste was first preheated in a heater (H, 135 °C) and was then transferred along with pressurised hydrogen operating at 375 °C and 6996 kPa into a high-temperature and high-pressure reactor to produce gas and liquid fuel. Similar to pyrolysis, the main product obtained was hydrocarbon fuel. The reactor was set up based on the experimental results of Ding et al. [37]. The hydrocracking reaction involves the breaking of carbon–carbon bonds and the simultaneous addition of hydrogen, which makes the reaction complex. The products formed are uncertain, which makes it difficult to obtain accurate reaction kinetics. Hence, again, an RYield block was used instead, similar to pyrolysis simulation. 

The products formed were first passed through separator S-1 in order to remove any solids, while the rest of the stream was passed through a series of coolers (C-1 and C-2) to cool it to 15 °C, as it was not expected to form a high number of heavier compounds. The cooled products were separated as gaseous products from the top and liquids from the bottom in the second separator, S-2. The products were separated in order to obtain different cuts of hydrocarbon fuel for comparison with pyrolysis fuel. The mass and energy balance were estimated using Aspen Plus simulation. The operational parameters and design specifications used for modelling the process in Aspen Plus are shown in Table 12.

### 3.3. Life Cycle Assessment

Life cycle assessment (LCA) is a standard method that is used to evaluate environmental impacts during the life cycle of a product. Generally, LCA is carried out according to ISO standards framework ISO 14040 2006a and 2006b [66,67]. LCA is widely used as a source to compare and analyse different waste-management product systems in terms of their environmental effects [68]. 

#### 3.3.1. Goal and Scope

Goal and scope define the reasons to carry out the study and the target audience for the study. They also state what role LCA would play in the assessment. The criteria, interpretations, system boundaries, impact categories, and limitations are expected to be a part of a goal and scope [67]. The goal of LCA is to evaluate, compare, and analyse different plastic management types. Two recycling methods were analysed for environmental impacts, considering the potential alternatives for the chemical recycling of plastic waste: pyrolysis of HDPE and hydrocracking of HDPE. 

The functional unit is the chemical recycling of HDPE plastic waste plant with the conversion capacity of 1000 kg/h (8400 tons per year when estimating that the plant is operated for 8400 h per year). It is a generation-based function unit, based on the products generated in the functional unit at a particular time. The products produced in both hydrocracking and pyrolysis were considered hydrocarbon fuels based on the literature study [69] and simulation results. LCA is performed based upon a gate-to-gate approach, where the impact of transportation to the facility was not considered.

The system considered in the study was assumed to be based in the United Kingdom. The results obtained in the study can be utilized in other parts of the world, as the technologies considered are common and generic. Figure 18 represents the scope and the system boundary of LCA for the current study.

#### 3.3.2. Life Cycle Inventory

The inventory analysis includes collecting input and output data primarily based on material and energy flows within the product system. The major task includes listing flows, data collection, and calculation techniques, verifying the data, and refining the system boundaries [67].

In the pyrolysis process, HDPE waste plastic was heated in the absence of oxygen in order to break polymers into saturated hydrocarbon vapours. The condensable gases were converted into liquid and separated as pyrolysis oil. Process data for pyrolysis, considering a functional unit with a capacity of 1000 kg/h, are summarised in Table 13. Pyrolysis oil produced in the process was considered naphtha. Heavy hydrocarbon, which can be used as an alternative fuel for machinery, was assumed as heavy oil. Pyrolysis gas, the non-condensable part of the process, was assumed as natural gas with the heat duty from simulation results. The process also produces char as a coproduct, which can be used as charcoal. Table 13 also presented the unit value of impact for all inputs and avoided impact of outputs in terms of GWP100a for better understanding. 

In the case of hydrocracking, the products obtained were separated into hydrocarbon fuel and gaseous products. The properties of liquid fuel obtained resembled those of gasoline-range fuels, so it is assumed that unleaded gasoline is used in heavy machinery. On the other hand, based on the simulation results, the gaseous products obtained were used as a source of natural gas. The inventory data for the hydrocracking process are summarised in Table 14. In both scenarios, the utility used to power the process was assumed as the electricity supply provided in Great Britain from the SimaPro Ecoinvent database. The inputs and outputs of pyrolysis and hydrocracking were taken as the inventory data based on the process simulations performed for the two processes. Similarly, Table 14 also presented the unit value of impact for all inputs and the avoided impact of outputs in terms of GWP100a for better understanding. 

## 4. Conclusions

A comparative analysis of the pyrolysis and hydrocracking of HDPE waste plastic was investigated using process simulation and lifecycle assessment. Both processes were simulated using Aspen Plus software based on the available literature data and on optimum operating conditions. The simulation was also validated based on the principle of mass and energy conservation. In pyrolysis, the product produced was mainly oil with an excellent HHV of 44 MJ/kg, whereas for hydrocracking, its value was ~45 MJ/kg. Similarly, gaseous products were obtained in both cases, with the gas from hydrocracking resembling the properties of natural gas. Both the pyrolysis and hydrocracking as a method for the chemical recycling of HDPE plastic waste into liquid fuels presented a sustainable and promising route with energy benefits. Moreover, SimaPro software was utilized to compare the environmental impacts of both chemical recycling methods using a CML-IA baseline method. The inventory data were generated based on the process simulation results, and missing data were taken from the literature. Comparative analysis showed better performance from the hydrocracking in terms of the environmental impacts of abiotic depletion, abiotic depletion of fossil fuels, GWP100a, human toxicity, FAETP, marine aquatic ecotoxicity, TAETP, photochemical oxidation potential, acidification, and eutrophication potential compared pyrolysis.

## Figures and Tables

**Figure 1 molecules-27-08084-f001:**
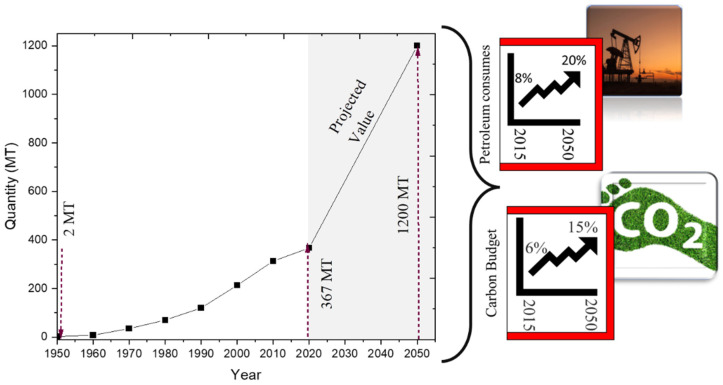
Global plastic production and its cost to the planet (MT = million metric ton) [4].

**Figure 2 molecules-27-08084-f002:**
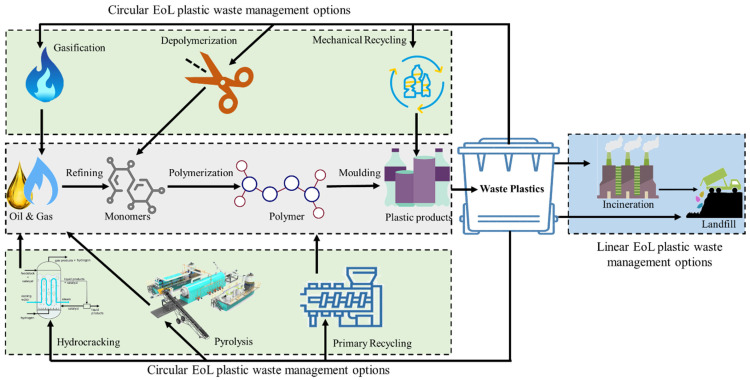
A comparative analysis of linear and circular EOL waste plastics management techniques.

**Figure 3 molecules-27-08084-f003:**
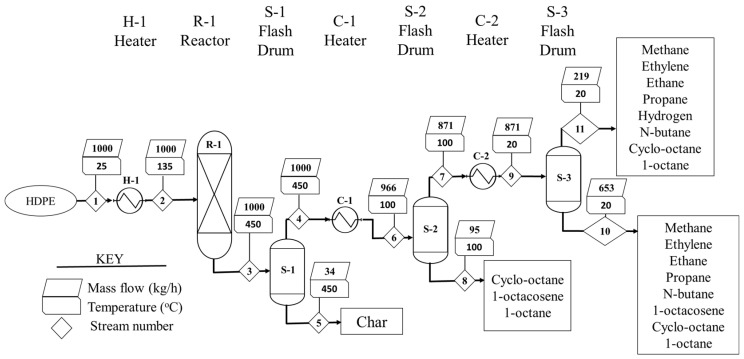
Process flow diagram and stream data for the pyrolysis process.

**Figure 4 molecules-27-08084-f004:**
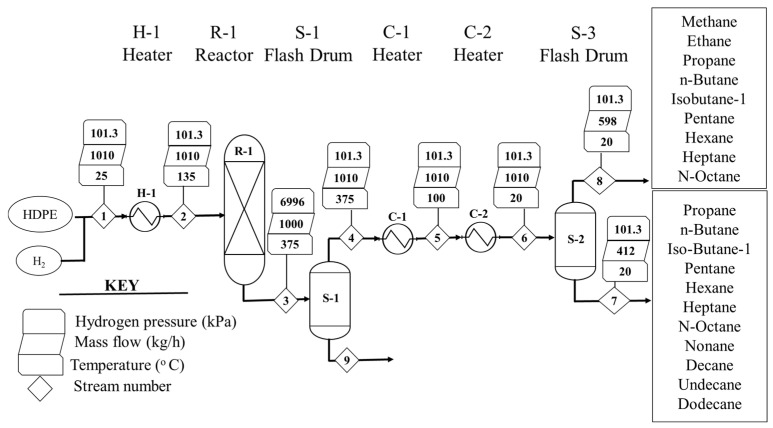
Process flow diagram and stream data for the hydrocracking process.

**Figure 5 molecules-27-08084-f005:**
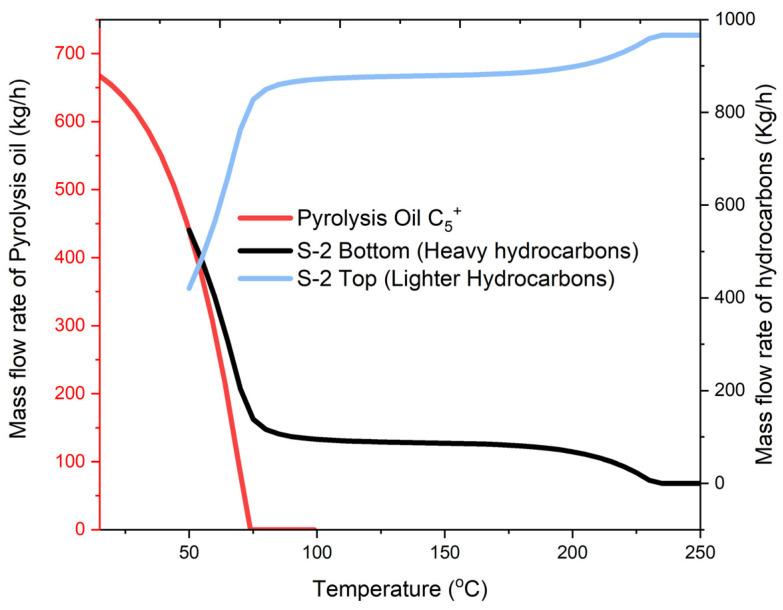
Sensitivity and optimization analysis of the pyrolysis process.

**Figure 6 molecules-27-08084-f006:**
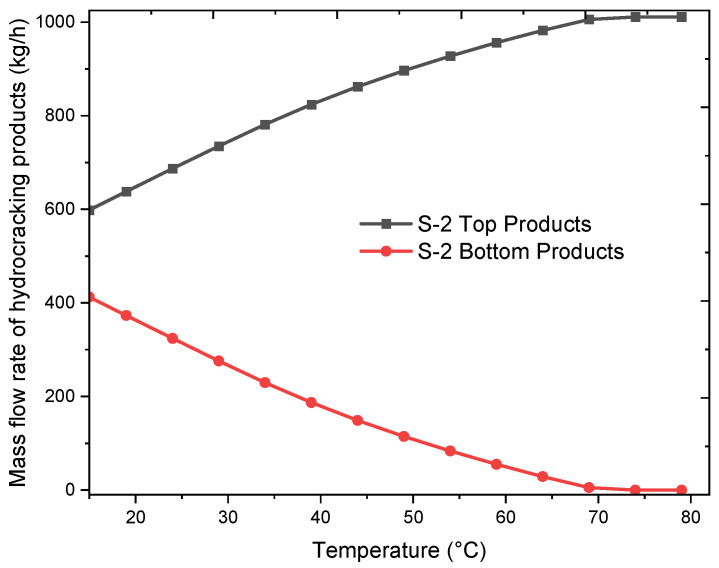
Sensitivity and optimization analysis of the hydrocracking process.

**Figure 7 molecules-27-08084-f007:**
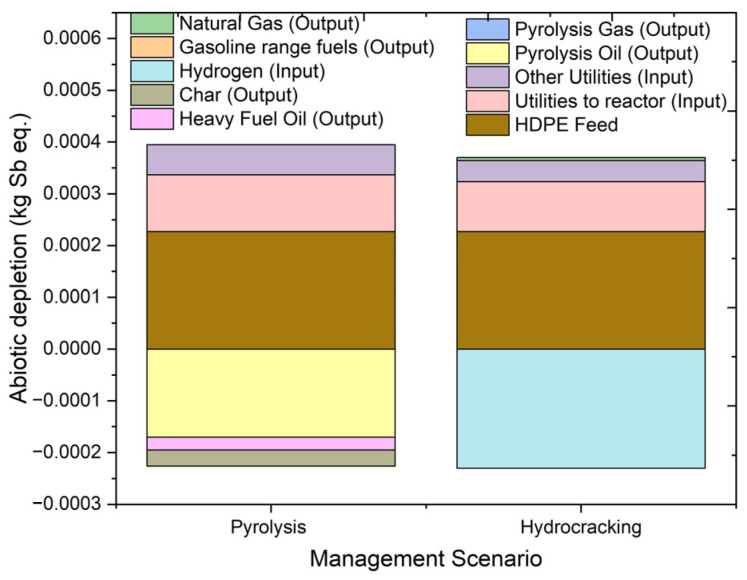
Impact assessment of pyrolysis and hydrocracking for the management of waste HDPE on depletion of abiotic resources (kg Sb eq.).

**Figure 8 molecules-27-08084-f008:**
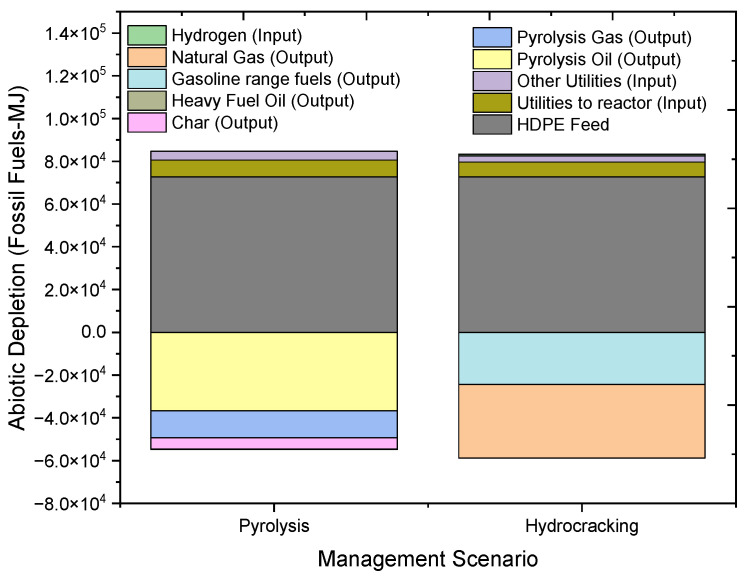
Impact assessment of pyrolysis and hydrocracking for the management of waste HDPE on abiotic depletion of fossil fuels (MJ).

**Figure 9 molecules-27-08084-f009:**
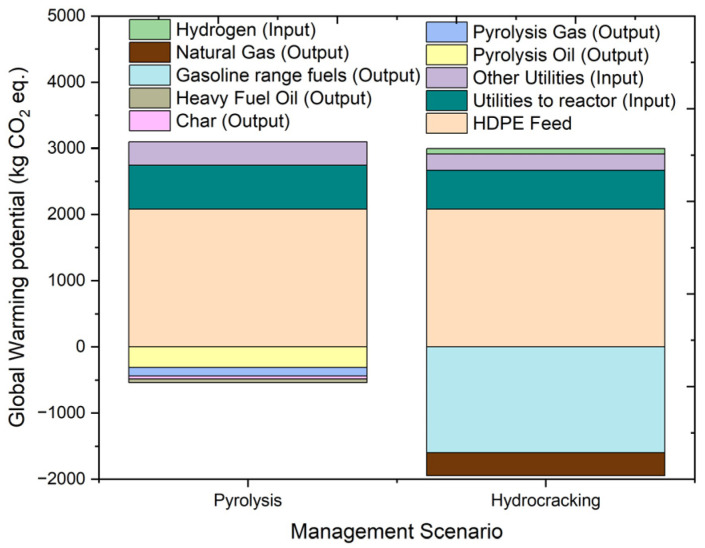
Impact assessment of pyrolysis and hydrocracking for the management of waste on global warming potential (kg CO_2_ eq.).

**Figure 10 molecules-27-08084-f010:**
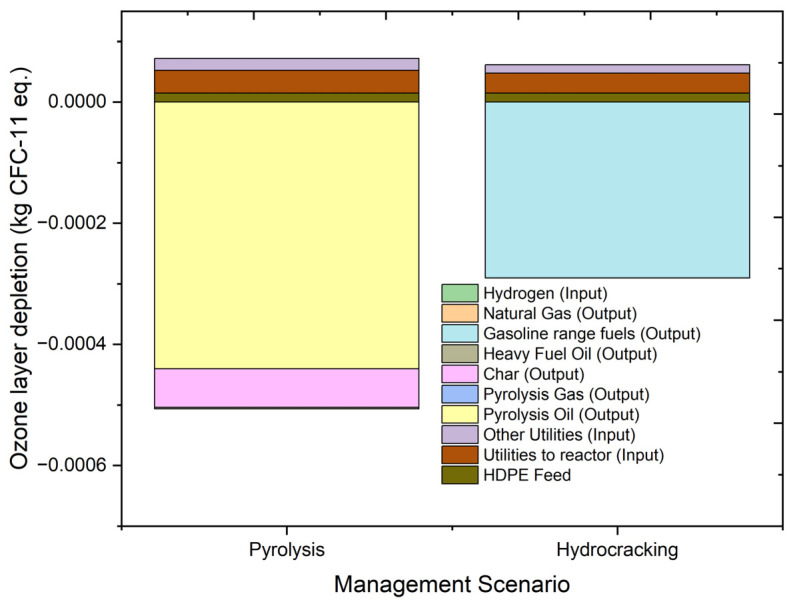
Impact assessment of pyrolysis and hydrocracking for the management of waste HDPE on ozone layer depletion (kg CFC-11 eq.).

**Figure 11 molecules-27-08084-f011:**
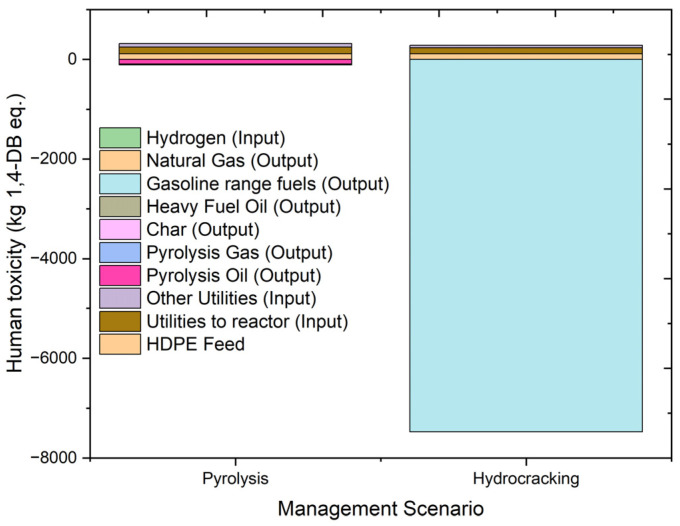
Impact assessment of pyrolysis and hydrocracking for the management of waste HDPE on human toxicity (kg 1,4-DB eq.).

**Figure 12 molecules-27-08084-f012:**
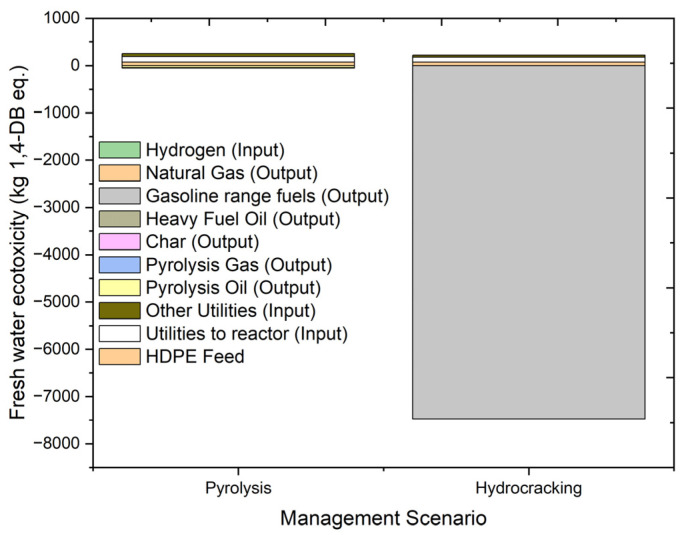
Impact assessment of pyrolysis and hydrocracking for the management of waste HDPE on fresh water ecotoxicity (kg 1,4-DB eq.).

**Figure 13 molecules-27-08084-f013:**
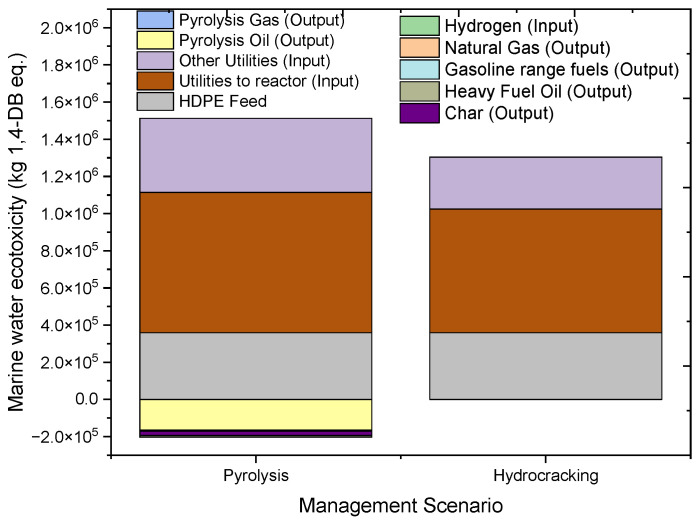
Impact assessment of pyrolysis and hydrocracking for the management of waste HDPE on marine aquatic ecotoxicity (kg 1,4-DB eq.).

**Figure 14 molecules-27-08084-f014:**
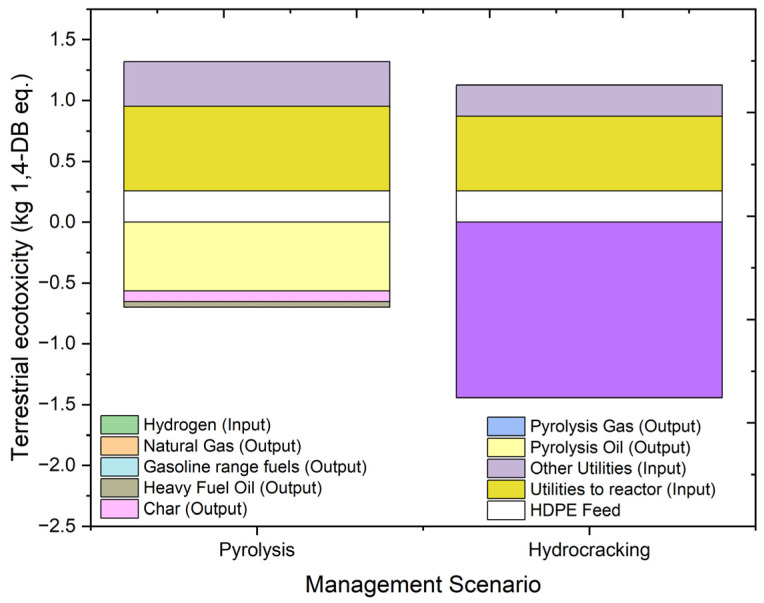
Impact assessment of pyrolysis and hydrocracking for the management of waste HDPE on terrestrial ecotoxicity (kg 1,4-DB eq.).

**Figure 15 molecules-27-08084-f015:**
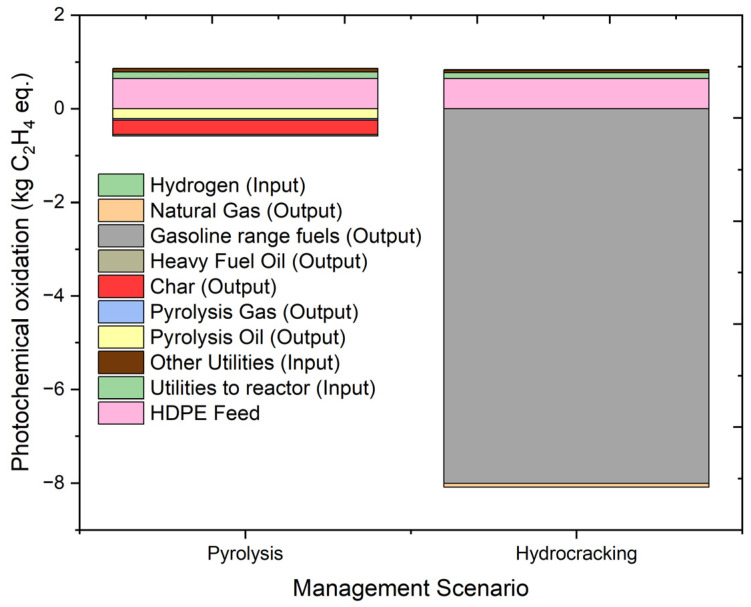
Impact assessment of pyrolysis and hydrocracking for the management of waste HDPE on photochemical oxidation (kg C_2_H_4_ eq.).

**Figure 16 molecules-27-08084-f016:**
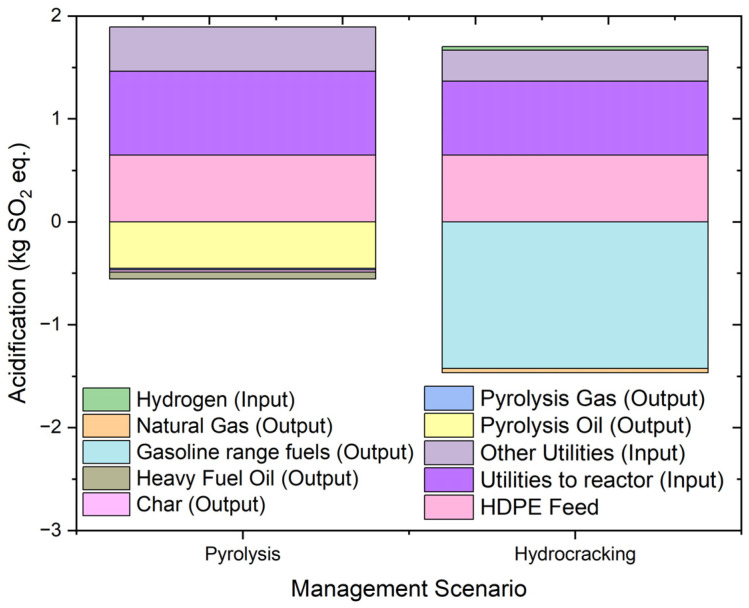
Impact assessment of pyrolysis and hydrocracking for the management of waste HDPE on acidification potential (kg SO_2_ eq.).

**Figure 17 molecules-27-08084-f017:**
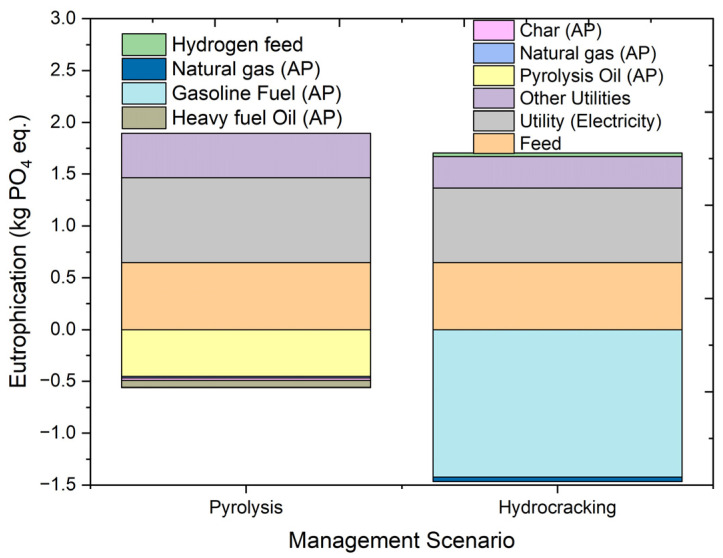
Impact assessment of pyrolysis and hydrocracking for the management of waste HDPE on eutrophication potential (kg PO_4_ eq.).

**Figure 18 molecules-27-08084-f018:**
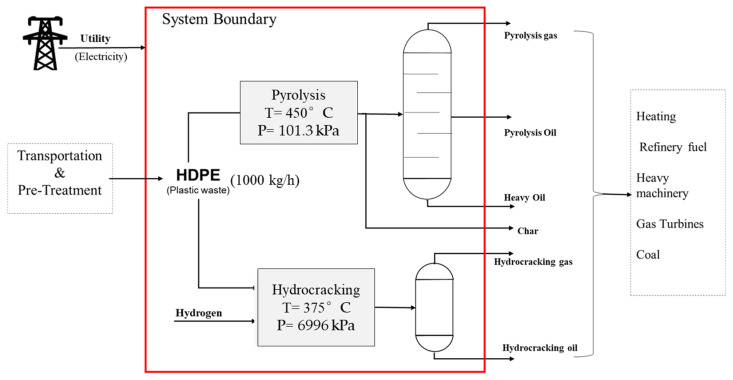
System boundary for the lifecycle assessment of pyrolysis and hydrocracking scenarios.

**Table 1 molecules-27-08084-t001:** Stream data for pyrolysis process.

Stream	Units	1	2	3	4	5	6	7	8	9	10	11
From	°	HDPE	H-1	R-1	S-1	S-1	C-1	S-2	S-2	C-2	S-3	S-3
To		H-1	R-1	S-1	C-1	Bottom	S-2	C-2	Bottom	S-3	Bottom	Top
T	°C	25	135	450	450	450	100	100	100	20	20	20
P	kPa	101.3	101.3	101.3	101.3	101.3	101.3	101.3	101.3	101.3	101.3	101.3
Mass Flow	kg/h	1000	1000	1000	966.22	33.77	966.22	871.26	94.97	871.26	652.54	218.71
Mass flow of HDPE	kg/h	1000	1000	0	0	0	0	0	0	0	0	0
Methane	kg/h	0	0	43.4	43.4	0	43.4	43.4	0	43.4	0.03	43.37
Ethylene	kg/h	0	0	6.68	6.68	0	6.68	6.68	0	6.68	0.02	6.66
Ethane	kg/h	0	0	22.7	22.7	0	22.7	22.7	0	22.7	0.09	22.61
Propane	kg/h	0	0	0.8	0.8	0	0.8	0.8	0	0.8	0.01	0.79
Hydrogen	kg/h	0	0	86.8	86.8	0	86.8	86.8	0	86.8	0	86.8
n-butane	Kg/h	0	0	0.42	0.42	0	0.42	0.42	0	0.42	0.02	0.4
Cyclooctane	kg/h	0	0	401.93	401.93	0	401.93	396	5.93	396	373.7	22.3
1-octacosene	kg/h	0	0	85.5	85.5	0	85.5	0.01	85.5	0.01	0	0
Char	kg/h	0	0	33.78	0	33.77	0	0	0	0	0.01	0
1-octane	kg/h	0	0	317.99	317.99	0	317.99	314.45	3.54	314.45	278.66	35.79
LHV	MJ/kg	-	-	50.55	51.17	32.79	51.17	52.04	43.15	52.04	44.06	75.86
H	kW	−1463.7	-	-	-	−274.84	-	-	−37.56	-	−323.66	−92.26

**Table 2 molecules-27-08084-t002:** Utility data for the pyrolysis process.

Equipment	Utility	Value	Heat Duty (kW)
H-1 (Preheater)	Medium Pressure steam (Inlet Temp: 175 °C, Outlet temp: 174 °C)	186.24 kg/h	105.27
R-1 (Reactor)	Electricity (US-EPA-Rule-E9-5711)	1096.68 kW	1096.68
C-1 (Cooler-1)	Cooling water (Inlet Temp: 20 °C, Outlet temp: 25 °C)	60,476.5 kg/h	−351.08
C-2 (Cooler-2)	Refrigerant 1 (Inlet Temp: −25 °C, Outlet temp: −24 °C)	109,325 kg/h	−121.47

**Table 3 molecules-27-08084-t003:** Stream data for the hydrocracking process.

	Units	1	2	3	4	5	6	7	8
From			H-1	R-1	S-1	C-1	C-2	S-2	S-2
To		H-1	R-1	S-1	C-1	C-2	S-2	Bottom	Top
T	°C	25	135	375	375	80	15	15	15
P	kPa	101.3	101.3	6996	101.3	101.3	101.3	101.3	101.3
Mass flow	kg/h	1010	1010	1010	1010	1010	1010	412.04	597.96
PE	kg/h	1000	1000	0	0	0	0	0	0
Methane	kg/h	0	0	5.44	5.44	5.44	5.44	0.01	5.43
Ethane	kg/h	0	0	24.58	24.58	24.58	24.58	0.42	24.16
Propane	kg/h	0	0	211.13	211.13	211.13	211.13	12.74	198.39
Hydrogen	kg/h	10	10	0	0	0	0	0	0
n-Butane	kg/h	0	0	73.83	73.83	73.83	73.83	14.82	59.01
n-Octane	kg/h	0	0	50.26	50.26	50.26	50.26	48.92	1.34
Isobu-01	kg/h	0	0	230.67	230.67	230.67	230.67	34.29	196.38
Pentane	kg/h	0	0	110.20	110.20	110.20	110.20	51.91	58.29
Hexane	kg/h	0	0	218.18	218.18	218.18	218.18	166.14	52.04
Heptane	kg/h	0	0	30.72	30.72	30.72	30.72	28.17	2.55
Nonane	kg/h	0	0	38.28	38.28	38.28	38.28	37.95	0.33
Decane	kg/h	0	0	14.30	14.30	14.30	14.30	14.26	0.04
Undecane	kg/h	0	0	1.81	1.81	1.81	1.81	1.81	0
Dodecane	kg/h	0	0	0.60	0.60	0.60	0.60	0.60	0
LHV	MJ/kg	-	119.96	-	-	-	-	44.81	45.82
H	kW	−1463.7	5.01	-	-	-	-	−268.97	−385.50

**Table 4 molecules-27-08084-t004:** Utility data for the hydrocracking process.

Equipment	Utility	Value	Heat Duty (kW)
H-1 (Preheater)	Medium Pressure steam (Inlet Temp: 175 °C, Outlet temp: 174 °C)	186.24 kg/h	105.27
R-1 (Reactor)	Electricity (US-EPA-Rule-E9-5711)	965.71 kW/h	965.71
C-1 (Cooler-1)	Cooling water (Inlet Temp: 20 °C, Outlet temp: 25 °C)	36217.7 kg/h	−210.02
C-2 (Cooler-2)	Refrigerant 1 (Inlet Temp: −25 °C, Outlet temp: −24 °C)	6653.4 kg/h	−73.97

**Table 5 molecules-27-08084-t005:** Pyrolysis process product composition based on process simulation.

Product	Composition	Mass Flow Rate (kg/h)	CV (MJ/kg)	Mass %
Pyrolysis oil	C_5_–C_8_	652.54	44.06	65.25
Heavy oil (Pyrolysis)	C_20_^+^	94.97	43.15	9.5
Pyrolysis gas	C_1_–C_4_+ Hydrogen	218.71	75.86	21.8
Char	Carbon	33.77	32.79	3.37

**Table 6 molecules-27-08084-t006:** Hydrocracking process products’ composition based on process simulation.

Products	Main Composition	Mass Flow Rate (kg/h)	CV (MJ/kg)	Mass %
Liquid fuel	C_4_–C_12_	412.04	44.81	41.2
Gaseous Products	C_1_–C_5_	597.96	45.82	59.8

**Table 7 molecules-27-08084-t007:** Environmental impact due to pyrolysis and hydrocracking.

	Impact Assessment of Pyrolysis Scenario								
Impact Category	Unit	Total	Feed	Utility (Electricity)	Other Utilities	Pyrolysis Oil (AP)	Natural Gas (AP)	Char (AP)	Heavy Fuel Oil (AP)
AD	kg Sb eq.	0.00017	0.000227	0.00011	5.79 × 10^−5^	−0.00017	−1.1 × 10^−7^	−3.1 × 10^−5^	−2.5 × 10^−5^
AD (FF)	MJ	29,901.52	72,773.53	7799.18	4116.629	−36652.5	−12581.8	−259.8	−5293.71
GWP100a	kg CO_2_ eq.	2557.997	2079.367	667.0891	352.1086	−309.6	−127.437	−57.2218	−46.3102
ODP	kg CFC-11 eq.	−0.00044	1.48 × 10^−5^	3.75 × 10^−5^	1.98 × 10^−5^	−0.00044	0	−2.6 × 10^−6^	−6.4 × 10^−5^
Human toxicity	kg 1,4-DB eq.	208.9515	114.9086	134.73	71.11432	−91.9663	−0.63464	−5.39463	−13.8059
FAETP.	kg 1,4-DB eq.	204.0771	73.29645	119.5066	63.07899	−42.9163	−0.0007	−2.64205	−6.24587
MAETP	kg 1,4-DB eq.	1,309,682	358,565.9	755073.6	398549.3	−165002	−5941.07	−8967.17	−22,595.8
TAETP	kg 1,4-DB eq.	0.620252	0.255168	0.696081	0.367412	−0.56392	−0.00116	−0.04524	−0.08809
Photochemical oxidation	kg C_2_H_4_ eq.	0.281434	0.644181	0.142009	0.074957	−0.2152	−0.03066	−0.30283	−0.03103
Acidification	kg SO_2_ eq.	7.820713	7.030695	3.420981	1.805691	−3.53334	−0.30266	−0.09266	−0.50799
Eutrophication	kg PO_4_ eq.	1.336317	0.647215	0.815912	0.430661	−0.45225	−0.01586	−0.02251	−0.06685
	**Impact Assessment of Hydrocracking Scenario**								
**Impact Category**	**Unit**	**Total**	**Feed**	**Utility (Electricity)**	**Other Utilities**		**Gasoline Fuel (AP)**	**Natural Gas (AP)**	**Hydrogen Feed**
AD	kg Sb eq.	0.000143	0.000227	9.66 × 10^−5^	4.07 × 10^−5^		−0.00023	−2.9 × 10^−7^	5.94 × 10^−6^
AD (FF)	MJ	24,528.46	72,773.52	6871.96	2892.67		−24,380.7	−34,399	769.9084
GWP100a	kg CO_2_ eq.	1049.399	2079.367	587.781	247.4194		−1598.73	−348.416	81.97481
ODP	kg CFC-11 eq.	−0.00023	1.48 × 10^−5^	3.3 × 10^−5^	1.39 × 10^−5^		−0.00029	0	0
Human toxicity	kg 1,4-DB eq.	−7194.09	114.9086	118.7124	49.97057		−7476.35	−1.73513	0.411386
FAETP	kg 1,4-DB eq.	188.8607	73.29645	105.2988	44.32431		−34.0578	−0.00191	0.000792
MAETP	kg 1,4-DB eq.	1,156,427	358,565.9	665,305.2	280,052.4		−132,047	−16,243.1	793.1172
TAETP	kg 1,4-DB eq.	−0.31712	0.255168	0.613326	0.258173		−1.44141	−0.00316	0.000789
Photochemical oxidation	kg C_2_H_4_ eq.	−7.25243	0.644181	0.125126	0.05267		−8.00329	−0.08383	0.012712
Acidification	kg SO_2_ eq.	3.593768	7.030695	3.014272	1.268822		−7.10891	−0.82748	0.216369
Eutrophication	kg PO_4_ eq.	0.236964	0.647215	0.718911	0.302617		−1.42326	−0.04336	0.034842
AP = Avoided Products									

**Table 10 molecules-27-08084-t010:** Feedstock conditions for the pyrolysis and hydrocracking processes.

Parameter	Pyrolysis	Hydrocracking
ṁ_HDPE_	1000 kg/h	1000 kg/h
ṁ_H2_	0	10 kg/h
* T_F_	25 °C	25 °C
** ρ	967 kg/m^3^	967 kg/m^3^
*** T_R_	450 °C	375 °C
**** P_R_	100.1 kPa	6996 kPa

ṁ = mass flow rate; * T_F_ = feed temperature; ** ρ = density; *** T_R_ = reactor temperature; ***** P_R_ = reactor pressure.

**Table 11 molecules-27-08084-t011:** Process model for the pyrolysis of HDPE waste plastic.

Equipment	Description	Aspen Model
H-1	Temperature 135 °C Pressure 100.3 kPa	HEATER
R-1	Temperature 450 °C Pressure 101.3 kPa HDPE flow rate 1000 kg/h	RYield
S-1	Temperature 450 °C Pressure 100.3 kPa	FLASH 2
C -1	Temperature 100 °C Pressure 100.3 kPa	HEATER
S-2	Temperature 100 °C Pressure 100.3 kPa	FLASH 2
C-2	Temperature 20 °C Pressure 100.3 kPa	HEATER
S-3	Temperature 20 °C Pressure 100.3 kPa	FLASH 2
Thermodynamic package: Peng Robinson		

**Table 12 molecules-27-08084-t012:** Process model for hydrocracking of HDPE waste plastic.

Equipment	Description	Aspen Model
H-1	Temperature 135 °C Pressure 100.3 kPa	HEATER
R-1	Temperature 375 °C Pressure 6996 kPa HDPE flow rate 1000 kg/h	RYield
S-1	Temperature 375 °C Pressure 100.3 kPa	FLASH 2
C -1	Temperature 100 °C Pressure 100.3 kPa	HEATER
C-2	Temperature 15 °C Pressure 100.3 kPa	HEATER
S-2	Temperature 15 °C Pressure 100.3 kPa	FLASH 2
Thermodynamic package: Peng-Robinson		

**Table 13 molecules-27-08084-t013:** Inventory data for the pyrolysis process per functional unit based on the simulation results.

Input.					
No.		Quantity	Remarks		Respective GWP100a (kg CO_2_ eq.)
1	Electricity	1674.5 kW	Electricity, high voltage GB Data		0.609/kWh
2	HDPE	1000 kg/h	Polyethylene, high density, granulate		2.08/kg
**Output**					
		**Quantity**	**CV (MJ/kg)**	**Substitute**	**Avoided GWP100a (kg CO_2_ eq.)**
1	Gaseous Products	218.72 kg/h	75.86	Natural gas	−0.583/kg
2	Char	33.7 kg/h	32.79	Char coal	−1.69/kg
3	Pyrolysis Oil	652.55 kg/h	44.056	Naphtha, petroleum refinery operations	−0.474/kg
4	Heavy Oil	94.97 kg/h	43.15 MJ/kg	Heavy fuel oil	−0.488/kg

**Table 14 molecules-27-08084-t014:** Inventory for the hydrocracking process per functional unit based on the simulation results.

Input					
No		Quantity	Remarks		Respective GWP100a (kg CO_2_ eq.)
1	Electricity	1372.2 kW	Electricity, high voltage GB Data		0.609/kWh
2	Mass flow of HDPE	1000 kg/h	Polyethylene, high density, granulate		2.08/kg
3	Mass flow of H_2_	10 kg/h	Hydrogen reforming		8.19/kg
**Output**					
		**Quantity**	**CV (MJ/kg)**	**Substitution**	**Avoided GWP100a (kg CO_2_ eq.)**
4	Liquids	412 kg/h	44.81	Petrol, unleaded	−3.88/kg
5	Gaseous Products	598 kg/h	45.8	Natural gas E	−0.583/kg

## Data Availability

The data presented in this study are available on request from the corresponding author.

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
