# Peer review of "Process Simulation and Life Cycle Assessment of Waste Plastics: A Comparison of Pyrolysis and Hydrocracking"

_molecules, 2022, doi:10.3390/molecules27228084_

Round 1

Reviewer 1 Report

The manuscript simulated the pyrolysis and hydrocracking processes of high-density polyethylene through Aspen Plus and conducted life cycle assessment. The results show that the hydrocracking scenario is more advantageous than pyrolysis based on the environmental impacts. This study will help to improve plastic waste management. However, there are still many issues that should be considered.

1.     Figure 3 involved in 2.1.1 and 2.1.2 does not conform to the content of the manuscript.

2.     The mass flow in Table 1 is unbalanced, please check the data carefully.

3.     The data sources of product enthalpy should be added and listed in table.

4.     In section 2.2, the optimal temperature of C-2 is 100 ℃, which is inconsistent with the content before and after the manuscript, and the analysis of hydrocracking process optimization in this section is repeated with pyrolysis.

5.     In section 2.3, the yield of the product described in the manuscript is inconsistent with the data in Table 5 and Table 6.

6.     In section 2.4.1, the impact of coke on the environment is inconsistent with the data in Table 7. Why does pyrolysis have a lower impact in the category of fossil fuel depletion on the environment?

7.     In section 2.4.6, why coke creates a negative impact on marine ecosystem.

8.     In section 3.1, the literature review related to the effect of hydrogen pressure and time on the chemical recovery of HDPE should be added.

9.     Most of the figures in section 2.4 are inconsistent with the data in Table 7, please check carefully.

10.  The materials and methods section is usually preceded by the results and discussion section, which makes it easier for the reader to read the article.

Author Response

Dear Reviewer,

We are grateful for your careful consideration and insightful comments.

Now, we are submitting the revised manuscript entitled “Process simulation and life cycle assessment of waste plastics: a comparison of pyrolysis and hydrocracking”. We have seriously considered all your comments and suggestions and revised the manuscript accordingly. We are confident that the paper is now well-improved. We hope that you find our responses to the comments satisfactory. 

Thank you.

Waheed Afzal

Reviewer 2 Report

This paper is very useful for the evaluation of waste plastic recycling. I have some comments below;

1)

This paper reviews many reference papers, and introduces numerical data. However, it is unclear how to derive the initial data for the evaluation of this paper. Please explain Tables 1 to 4 in detail. The results and discussion in this paper depend on the initial data.

2)

In addition, this paper includes information from many references. It is important to review the literature related to this research, I think. However, many data have reference numbers. The large number of data presented from the references has diluted the results obtained in this study. I think it would be better to focus the data for the purpose of this research.

3)

This paper does not use symbols for the parameters. Tables and equations are shown in parameters only. Therefore, it is difficult to understand the relationship of each parameter. I recommend you to define the symbol for the parameters.

4)

The figures in this paper, for example Figs. 3 and 4, do not have a parameter on the vertical axis. Please show the parameter as well as the unit.

5)

This paper has many abbreviations. Please define them and make a nomenclature for them.

Author Response

(The authors gave the same response as above.)

Round 2

Reviewer 1 Report

The authors have addressed my concerns.

Reviewer 2 Report

This revised paper has been revised to reviewer comments. I agree to accept this paper for publication.